# Parallel Attention Network using Vector with High Correlation with Label for Remaining Useful Life Estimation

**Ye-In Park, Jouwon Song and Suk-Ju Kang**

Department of Electronic Engineering, Sogang University, Seoul, South Korea

## Abstract

Prognostic health management (PHM) has become important in many industries as a critical technology to increase machine stability and operational efficiency. Recently, various methods using deep learning to estimate the remaining useful life (RUL) as a core task of PHM have been proposed. However, the existing methods do not explicitly capture the correlation between temporal and spatial time series, reducing the RUL prediction accuracy. This paper proposes a novel RUL prediction algorithm using a spatio-temporal attention mechanism to based on the vector highly correlated with label to solve this problem. The proposed model constructs three paths in parallel, a time-oriented attention network, a feature-oriented attention network, and a bidirectional long short-term memory (LSTM) network. The first two attention networks focus on temporal and spatial information required for RUL prediction based on convolutional neural network (CNN), respectively. Unlike existing attention networks, the proposed attention network uses the vector learned in the intermediate prediction process as a query vector to focus on time series data related to the RUL. The last bidirectional LSTM network is additionally configured to compensate for the inability of the CNN-based attention networks to grasp continuous time distributions. Experiments have been performed on two widely used datasets and experimental results show that the proposed approach outperforms the state-of-the-arts.

## Introduction

Systems such as aircraft, space probes, nuclear power generators, and wind power generators require operational efficiency, high reliability, and high performance under extreme loads. Hence, prognostic health management (PHM) has become critical to improve them (Batzel and Swanson 2009). Specifically, in PHM, prognostics predict the remaining useful life (RUL) to determine whether a problem exists for the system to perform the intended function, which plays a decisive role in PHM (Wang et al. 2018). The importance of RUL prediction in many fields has encouraged researchers to develop various RUL prediction approaches.

Recently, the data-driven approach relies on historically collected data and attempts to derive models directly from the data for RUL prediction (Kim, An, and Choi 2017). The

data-driven approach has been becoming increasingly popular because no physical knowledge is required. In addition, the sensor system technology, data storage, and analysis technology have constantly improved, thereby increasing the use of the data-driven approach.

Deep learning has been actively used in the data-driven approach to develop high-performance RUL prediction algorithms (Huang, Huang, and Li 2019; Liu et al. 2020; Miao et al. 2019). Deep learning, which has remarkable performance in image and speech recognition, has a structure in which several layers are stacked to extract feature information from raw input data. This characteristic of deep learning has great potential for matching original data and the RUL, and several algorithms have been studied in this regard (Hsu and Jiang 2018; Wang et al. 2018; Huang, Huang, and Li 2019; Babu, Zhao, and Li 2016; Liu et al. 2019; Li, Li, and He 2019; Al-Dulaimi et al. 2019, 2020). However, it is difficult to predict RUL more accurately because these methods cannot identify correlation between time-series data and cannot identify time-series data deeply related to RUL.

Recently, methods using an attention mechanism have been studied to solve these problems. In particular, a self-attention technique (Vaswani et al. 2017) has been widely applied (Wang et al. 2020; Zhang, Song, and Li 2021; Liu and Wang 2021). This technique is used to determine the relationships between data within the same data set. In other words, it is to grasp the correlation between sequence data, and it is not designed in a structure that can directly analyze the correlation between sequence data and the RUL.

We propose a novel attention network guided by vector highly correlated with label inspired by (Devlin et al. 2018) to solve this problem. (Devlin et al. 2018) randomly initializes and uses a classification token (CLS) that summarizes important information of text classification. Similar to this work, we generate and use a vector with high correlation with the target RUL for RUL prediction. Unlike existing methods using the attention mechanism, the proposed attention network trains the vector with high correlation with the RUL through an intermediate prediction process, and uses it as a query in a scaled dot product attention process. Thus, the correlation between the input data and RUL can be directly grasped. In addition, according to the correlation degree between the input data and RUL, a weight can be given accordingly. Therefore, the proposed method can predict the

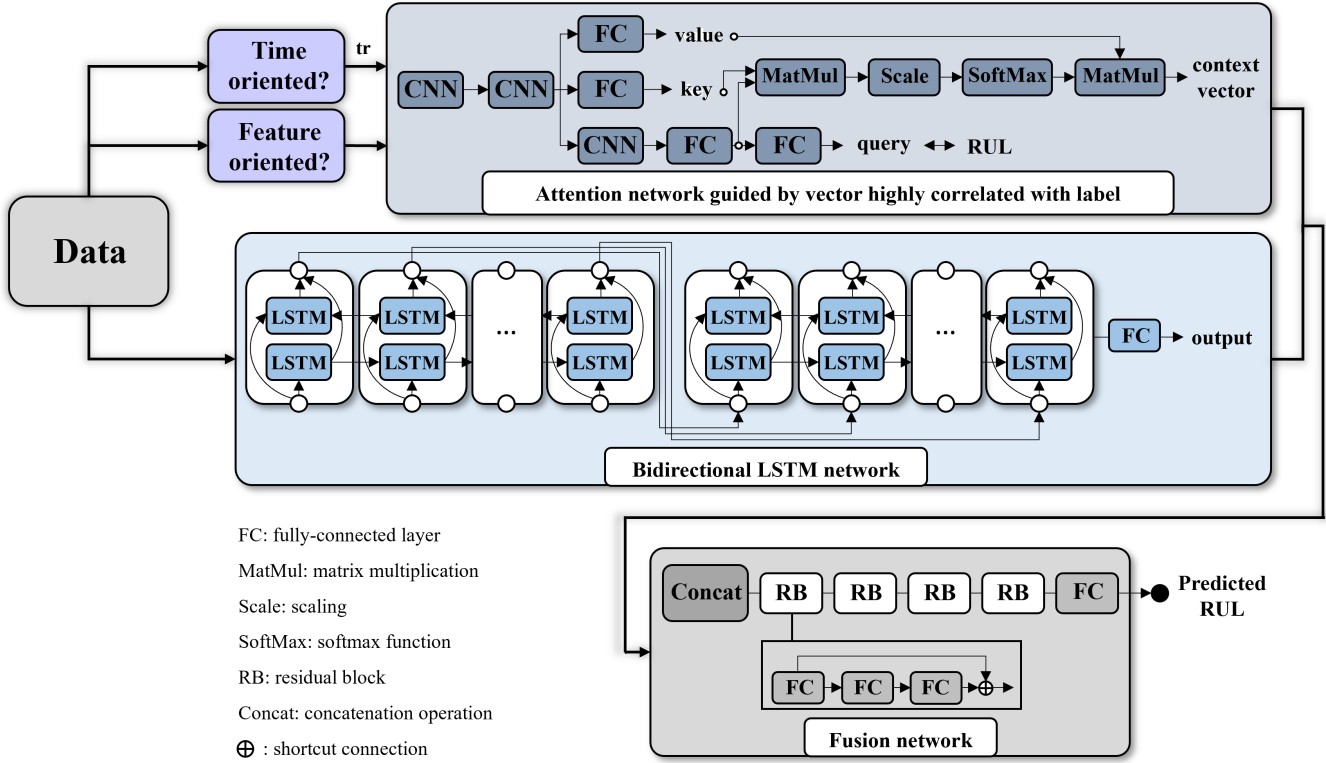

Figure 1: Overall architecture of the proposed method.

RUL more accurately than the existing methods.

## Proposed method

Fig. 1 shows the overall framework of the proposed method. The proposed network consists of four primary substructures: a time-oriented attention network guided by vector highly correlated with label, feature-oriented attention network guided by vector highly correlated with label, bidirectional LSTM network, and fusion network. The time-oriented attention network assigns weights according to the importance of each time step for the RUL prediction, and the feature-oriented attention network assigns weights according to the importance of each sensor for the RUL prediction. Both networks are CNN-based attention networks, which train a vector highly correlated with the RUL using the newly proposed intermediate prediction process and the vector as a query. Therefore, the attention networks provide higher weights to time steps or sensors that are deeply related to the RUL between time steps or sensors. The bidirectional LSTM network was additionally constructed to compensate for the fact that the two attention networks were CNN-based and could not grasp the continuous time distribution. The network is designed according to the temporal dimension and can store important historical data information by bidirectionally learning long-term dependencies between time steps in time series data. Finally, the fusion network applies the concatenation operation to the outputs of

the previous three path networks and outputs the final predicted RUL through residual blocks. The topology of the substructures is listed in Table 4, and the details of the substructures are described in the following sections.

## Attention Network guided by Vector Highly Correlated with Label

The overall structure of the proposed attention network is presented in the attention network guided by vector highly correlated with label block in Fig. 1. In this network, unlike the existing self-attention mechanism (Vaswani et al. 2017), an intermediate prediction process is newly designed in the middle of the attention network process. Through the intermediate prediction process, a vector with high correlation with the RUL is trained. Then, the learned vector is an input as a query to the scaled dot product attention process. Therefore, the key can determine the correlation with the query, which is a vector representing the RUL, so that the context vector, including the correlation information between the input data and RUL, can be obtained through the operation with a value. In addition, the proposed attention network is constructed in parallel in two dimensions: the spatial and temporal dimensions. Through this, the correlation between the temporal time series and RUL and between the spatial time series and RUL can be directly grasped, and all aspects of multivariate data are considered. Therefore, it was robust to the data set characteristics and exhibited high per-

formance. Furthermore, the proposed attention network provides excellent interpretability to easily determine the time series affecting the RUL through the attention score calculation. Details are presented in the results section.

The proposed attention network consists of two paths. One is a time-oriented attention network guided by vector highly correlated with label, and the other is a feature-oriented attention network guided by vector highly correlated with label. In the time-oriented attention network guided by vector highly correlated with label, a deeper RUL-related time step is assigned a greater weight in the corresponding time step. The attention network considers the sensor signal a channel in the CNN operation, extracts the features of each time step, and concentrates on the correlation between time steps. The original data shape is a vector in the spatial dimension based on the order of batch size, time, and sensor; thus, the input data shape is changed to the batch size, sensor, and time by transposing the row representing the time and the column representing the sensor. In the feature-oriented attention network guided by vector highly correlated with label, a deeper RUL among the sensors is assigned a greater weight for the corresponding sensor. The attention network considers the time signal to be a channel in the CNN operation, extracts the features of each sensor, and concentrates on the correlation between the sensors. The original data shape is an input to the network as is. Two attention networks have the same structure, as depicted in the attention network guided by vector highly correlated with label block in Fig. 1.

### Bidirectional LSTM Network

The bidirectional LSTM is used to capture long-term dependencies in the forward and backward directions of the input sequence data. The LSTM is a network developed to solve the vanishing gradient problem of the RNN, and the network considering the direction information in the LSTM is the bidirectional LSTM; hence, the bidirectional LSTM can capture long-term dependencies in both directions, preserving more information. In addition, the proposed attention network is CNN-based and does not consider temporal information between data. Thus, the bidirectional LSTM is used as one of the three-path parallel networks to use temporal information that cannot be handled by the attention network.

### Fusion network

The fusion network concatenates the outputs of the three path parallel networks and outputs the predicted RUL value, the final output. It consists of the concatenation operation, four residual blocks, and one fully connected layer, as illustrated in the fusion network block in Fig. 1. In the residual block, the process of connecting the previous layer through a shortcut connection (He et al. 2016) is applied. Through this, the operation is straightforward and fast, but it greatly affects solving the gradient vanishing and exploding problems by passing the gradient directly. Finally, the fully connected layer outputs the predicted RUL, the final output, by assigning one as the output feature size.

## Experiments

### Data Description and Processing

For the performance evaluation of the proposed method, the commonly used NASA Commercial Modular Aero-Propulsion System Simulation (C-MAPSS) dataset was used. The C-MAPSS dataset describes the degradation process of the aircraft engine. The engine consists of a fan, low-pressure compressor, high-pressure compressor, combustor, low-pressure turbine, and high-pressure turbine. A total of 21 on board sensors that measure temperature, pressure, and speed are placed in different locations to monitor engine conditions. The measured values of the sensors were sampled to compose a dataset as a multivariate time series. Description of the data set is listed in Table 5.

Among the 21 sensors recorded in the C-MAPSS dataset, low-importance sensors with no significant change in time series values are included. Such data do not provide helpful information to predict the RUL and increase the network complexity (Liu et al. 2019; Li, Li, and He 2019). Therefore, the proposed method selected 14 sensors with indexes of 2, 3, 4, 7, 8, 9, 11, 12, 13, 14, 15, 17, 20, and 21 like the existing methods (Mo et al. 2020; Lim, Goh, and Tan 2016). We used the min-max normalization to provide a standard range for all sensor measurements. In addition, engine performance degradation in the initial stage was almost negligible in actual use (Zheng et al. 2017). Therefore, the maximum value of the RUL was limited to a constant value. The maximum value of the RUL was set to 125, and the window size was set to 30. Detailed information of the additional data set used is described in Appendix A, B and Table 6.

### Evaluation Metrics and setting

This paper used two objective evaluation metrics, the root mean square error (RMSE) and score function, for performance evaluation. The score function in (1) is an evaluation metric proposed by the authors of the C-MAPSS dataset. In the real world, late forecasting can cause severe damage to the engine unit. Thus, a characteristic of the score function is that it imposes more penalties on late predictions than on early predictions (Lim, Goh, and Tan 2016). The score function is as follows:

$$
S_i = \begin{cases} exp^{-\frac{E_i}{13}} - 1, & \text{if } E_i < 0, \\ exp^{\frac{E_i}{10}} - 1, & \text{otherwise,} \end{cases} \quad Score = \sum_{i=1}^{n} S_i. \quad (1)
$$

Detailed information of the evaluation metrics used is described in Appendix C.

The learning rate was set to 0.0005, and the learning rate was decreased by 0.3 every 40, 80, 140, and 160 epochs. The batch size was set to 512, and training was conducted for 170 epochs. The Adam optimizer was applied.

### Results

**Comparison with State-of-the-arts Methods**  Several conventional methods (Babu, Zhao, and Li 2016; Hsu and Jiang 2018; Wang et al. 2018; Liu et al. 2019; Li, Li, and He 2019; Mo et al. 2020; Li et al. 2020; Al-Dulaimi et al. 2020;

Table 1: Performance comparison of the proposed and state-of-the-art methods on the C-MAPSS dataset (RMSE / Score)

| Method | FD001 | FD002 | FD003 | FD004 |
|---|---|---|---|---|
| Babu et al. 2016 | 20.74/973 | 23.53/3184 | 19.21/745 | 28.51/5867 |
| Hsu et al. 2018 | 14.05/281 | 16.66/1005 | 14.08/258 | 20.61/2379 |
| Wang et al. 2018 | 13.89/271 | 15.87/985 | 13.00/189 | 19.78/2360 |
| Liu et al. 2019 | 15.63/526 | 21.03/3182 | 14.27/324 | 21.77/4682 |
| Li et al. 2019 | 20.41/1058 | 20.24/2336 | 13.18/335 | 23.29/6749 |
| Mo et al. 2020 | 12.19/259 | 19.93/4350 | 12.85/343 | 22.89/4340 |
| Li et al. 2020 | 11.44/196 | 19.35/3747 | 11.67/242 | 22.22/4844 |
| Al- et al. 2020 | 12.32/238 | 15.04/1057 | 11.36/226 | 17.75/1357 |
| Wang et al. 2019 | 10.95/261 | 20.47/4368 | 10.62/247 | 22.64/5168 |
| Huang et al. 2019 | - | 25.11/4793 | - | 26.61/4971 |
| Proposed | 11.25/183 | 14.96/899 | 10.81/152 | 16.17/1588 |

Table 2: Performance comparison of variants of the proposed model on the C-MAPSS dataset (A): Feature-oriented attention network guided by vector highly correlated with label (B): Time-oriented attention network guided by vector highly correlated with label (RMSE / Score)

| Method | FD001 | FD002 | FD003 | FD004 |
|---|---|---|---|---|
| (A) removal | 11.98/206 | 15.48/935 | 12.12/194 | 17.79/1706 |
| (B) removal | 12.82/242 | 15.52/931 | 10.76/151 | 17.65/1719 |
| (A),(B) removal | 12.77/254 | 15.98/974 | 11.59/175 | 18.71/1818 |
| Proposed | 11.25/183 | 14.96/899 | 10.81/152 | 16.17/1588 |

Table 3: Performance comparison of attention network guided by vector highly correlated with label and self-attention network on the C-MAPSS dataset (RMSE / Score)

| Method | FD001 | FD002 | FD003 | FD004 |
|---|---|---|---|---|
| Self-attention | 12.26/210 | 15.76/1094 | 12.17/204 | 18.50/2226 |
| Proposed | 11.25/183 | 14.96/899 | 10.81/152 | 16.17/1588 |

Wang et al. 2019; Huang, Huang, and Li 2019) were used for performance comparison with the proposed method. As listed in Table 1, the proposed method achieved higher performance in FD001, FD002, FD003, and FD004 for the RMSE, on average, 2.95, 6.28, 3.04, and 7.4 than the state-of-the-art methods on the C-MAPSS dataset, respectively. In addition, the proposed method achieved higher performance in FD001, FD002, FD003, and FD004 for the score than the state-of-the-art methods, on average, 198, 2311, 245, and 3299, respectively. (Wang et al. 2019) had 0.3 and 0.19 higher RMSE performance than the proposed method in FD001 and FD003, respectively, but 5.51 and 6.47 lower RMSE performance in FD002 and FD004 than the proposed method, respectively. Furthermore, the existing methods had significantly poor performance in FD002 and FD004 compared to FD001 and FD003, but the proposed model has proved to be the most robust model for the data set characteristics by remarkably narrowing the performance gap. Performance comparisons for the additional data sets used are described in Appendix D and Table 7.

**Ablation study**

The ablation study was performed to remove the applied techniques from the proposed method to confirm the effects on the C-MAPSS dataset. The ablation study was performed to remove the applied techniques from the proposed method to confirm the effects on the C-MAPSS dataset. First, to confirm the effect of using both the time- and feature-oriented networks, we compared the variant performance in which each attention network was removed. Second, to verify the effectiveness of the newly proposed attention network, an attention network guided by vector highly correlated with label, we changed the proposed attention network to an existing self-attention network and compared the performance. Specific details are explained in the section below.

**Effectiveness of time- and feature-oriented attention networks** We compared the variant performance in which each attention network is removed to check the effectiveness of using time and feature-oriented networks. In Table 2, (A) and (B) denote feature- and time-oriented networks,

respectively. The proposed method achieved better performance than all other variants based on the RMSE. Through this, we prove the effectiveness of using both the time- and feature-oriented networks.

**Effectiveness of attention network guided by vector highly correlated with label** To verify the effectiveness of the newly proposed attention network, we changed it to an existing self-attention network and compared their performance. Table 3 reveals that the proposed method achieved better performance for the RMSE and score function than the self-attention method. Through this, we prove that the attention network has an effect comparable to the self-attention method that has been widely used, and we prove the need for an intermediate prediction process that predicts the label RUL using a query in the middle of the network.

## Conclusion

We proposed a novel attention network guided by vector highly correlated with label. The proposed attention network trained a query to highly correlate with the RUL through an intermediate prediction process. Then, the learned query was applied to the scaled dot product attention process. Thus, the correlation between the input data and RUL was directly identified, and weights were assigned accordingly. In addition, data related to the RUL were identified in both dimensions by constructing the proposed attention network in parallel in the temporal and spatial dimensions. Therefore, the correlation between the temporal time series and RUL and between the spatial time series and RUL is directly grasped, and a more accurate RUL prediction was possible in a form that considers all aspects of the multivariate data. In the experimental results, the proposed model outperformed state-of-the-art models for RUL prediction on various benchmark datasets.

## Acknowledgments

This research was supported by the MSIT(Ministry of Science and ICT), Korea, under the ITRC(Information Technology Research Center) support program(IITP-2021-2018-0-01421) supervised by the IITP(Institute of Information communications Technology Planning Evaluation), and was supported by the National Research Foundation of Korea (NRF) grant funded by the Korea government (MSIT) (No. 2021R1A2C1004208).

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

# Appendix A

## Data Description

To provide additional quantitative results of the proposed method, the IEEE PHM 2012 Prognostic Challenge data set (Nectoux et al. 2012) was used. In the IEEE PHM 2012 Prognostic challenge dataset, temperature and vibration data in the horizontal and vertical directions in the bearing housing were sampled at 25.6 kHz, and the process of stopping for 10 seconds was repeatedly performed and recorded. As presented in Table 6, this dataset includes three sub-datasets measured under different conditions. We adopted all sub-datasets for evaluation.

# Appendix B

## Data Processing

In the C-MAPSS dataset, the input sequence data can be expressed as a matrix, as shown in (2):

$$X = [x_1, x_2, ..., x_t, ..., x_T] = [x^1, x^2, ..., x^s, ..., x^S] \in R^{T \times S},$$
$$x_t = x_t^1, x_t^2, ..., x_t^S \in R^{1 \times S}, \ t \in [1, T],$$
$$x^s = x_1^s, x_2^s, ..., x_T^s \in R^{T \times 1}, \ s \in [1, S], \quad (2)$$

where t and T represent the time-sequential index and final failure time, respectively. Moreover, s and S represent the sensor index and number of sensors, respectively.

In the IEEE PHM 2012 Prognostic challenge dataset, the input sequence data can be expressed as a matrix, as in (3):

$$X = [x_1, x_2, ..., x_t, ..., x_T] \in R^T,$$
$$x_t = x_t^1, ..., x_t^v, ..., x_t^V \in R^{1 \times V}, \ t \in [1, T], \quad (3)$$

where $t$ and $T$ represent the time-sequential index and final failure time, respectively. In addition, $v$ and $V$ represent the vibration signal type index and number of vibration signal types, respectively. There are two types of vibration signals: horizontal and vertical. As with existing methods (Mao et al. 2018; Soualhi, Medjaher, and Zerhouni 2014), we performed the RUL prediction on univariate data using only the horizontal signal because it provides better results for tracking the bearing degradation. In addition, standardization was used to consider values that deviate significantly from the mean as outliers. The RUL was scaled using a health indicator (HI) (Yang et al. 2020) as in (4):

$$HI_t = \frac{RUL_t}{RUL_0} \quad (4)$$

where $HI$ at time $t$ can be obtained by dividing the RUL at $t$ ($RUL_t$) by the RUL at the origin ($RUL_0$). A series of $HI$ values ranging from 0 to 1 is obtained.

# Appendix C

## Evaluation Metrics

For performance evaluation, two objective evaluation metrics were used: the RMSE and score function. The RMSE not described above is defined as follows:

$$E_i = \overline{RUL_i} - RUL_i, \ RMSE = \sqrt{\frac{1}{n} \sum_{i=1}^{n} E_i^2}, \quad (5)$$

where $\overline{RUL_i}$ and $RUL_i$ are the predicted RUL and true RUL, respectively, $n$ represents the total number of test samples, and $E_i$ denotes the difference between the predicted and actual values of the $i^{th}$ test sample.

The score function in (6) is an evaluation metric proposed by the authors of the IEEE PHM 2012 Prognostic challenge dataset. Similar to (1), it is characterized by imposing more penalties on late prediction values than on early prediction values. The score function is as follows:

$$Score = \frac{1}{n} \sum_{i=1}^{n} A_i,$$

$$A_i = \begin{cases} exp\left(-ln(0.5)\left(\frac{Er_i}{5}\right)\right), & Er_i \leq 0, \\ exp\left(-ln(0.5)\left(\frac{Er_i}{20}\right)\right), & Er_i > 0, \end{cases}$$

$$Er_i = \frac{\overline{RUL_i} - RUL_i}{RUL_i} \times 100. \quad (6)$$

# Appendix D

## Quantitative Results on the IEEE PHM 2012 Prognostic challenge dataset

We performed experiments on the IEEE PHM 2012 Prognostic Challenge dataset to provide additional quantitative results of the proposed method, as shown in Table 7. In RMSE, the proposed method showed higher performance than the existing method with an average of 4485, 3435, and 1199 of conditions 1, 2, and 3, respectively. In addition, in score function, the proposed method showed higher performance than the existing method with an average of 199.93, 150.49, and 146.19 of conditions 1, 2, and 3, respectively. In particular, in conditions 1 and 2, the lowest performances based on RMSE were 27887 and 20635 by (Hsu and Jiang 2018), respectively. However, for the same sub-datasets, the proposed method showed 2360 and 1099, respectively, and had a big performance difference of 25527 and 19536, respectively, compared to (Hsu and Jiang 2018). We prove the

superiority of the proposed method in both datasets through performance comparison on the IEEE PHM 2012 Prognostic Challenge dataset as well as the performance comparison on the C-MAPSS dataset shown above.

Table 4: Topology of the overall architecture

| Layer | Parameters | Input channel size | Output channel size |
|---|---|---|---|
| Time-oriented attention network | | | |
| CNN_1 | kernel: 4, stride: 1 | 14 | 100 |
| CNN_2 | kernel: 3, stride: 1 | 100 | 100 |
| FC_1 | - | 22 | 30 |
| FC_2 | - | 22 | 30 |
| CNN_3 | kernel: 3, stride: 1 | 100 | 100 |
| FC_3 | - | 19 | 100 |
| FC_4 | - | 100 | 1 |
| Feature-oriented attention network | | | |
| CNN_1 | kernel: 4, stride: 1 | 30 | 100 |
| CNN_2 | kernel: 3, stride: 1 | 100 | 100 |
| FC_1 | - | 6 | 14 |
| FC_2 | - | 6 | 14 |
| CNN_3 | kernel: 3, stride: 1 | 100 | 100 |
| FC_3 | - | 3 | 100 |
| FC_4 | - | 100 | 1 |
| Bidirectional LSTM network | | | |
| BLSTM_1 | hidden: 64, activation: tanh | 14 | 64 |
| BLSTM_2 | hidden: 64, activation: tanh | 64 | 64 |
| FC | - | 6 | 100 |
| Fusion network | | | |
| RB_1 | - | 300 | 200 |
| RB_2 | - | 200 | 200 |
| RB_3 | - | 200 | 400 |
| RB_4 | - | 400 | 400 |
| FC_1 in RB | activation: ReLU | input of RB | output of RB |
| FC_2 in RB | dropout: 0.2 | output of RB | output of RB |
| FC_3 in RB | activation: ReLU | output of RB | output of RB |
| FC | - | 400 | 1 |

Table 5: Description of C-MAPSS dataset

| | FD001 | FD002 | FD003 | FD004 |
|---|---|---|---|---|
| Training engine number | 100 | 260 | 100 | 248 |
| Testing engine number | 100 | 259 | 100 | 248 |
| Operational conditions | 1 | 6 | 1 | 6 |
| Fault modes | 1 | 1 | 2 | 2 |

Table 6: Description of IEEE 2012 Prognostic challenge dataset

|  | Operational condition 1 | Operational condition 2 | Operational condition 3 |
|---|---|---|---|
| Load (N) | 4000 | 4200 | 5000 |
| Speed (rpm) | 1800 | 1650 | 1500 |
| Training dataset | Bearing 1_1 | Bearing 2_1 | Bearing 3_1 |
|  | Bearing 1_2 | Bearing 2_2 | Bearing 3_2 |
| Testing dataset | Bearing 1_3 | Bearing 2_3 | Bearing 3_3 |
|  | Bearing 1_4 | Bearing 2_4 |  |
|  | Bearing 1_5 | Bearing 2_5 |  |
|  | Bearing 1_6 | Bearing 2_6 |  |
|  | Bearing 1_7 | Bearing 2_7 |  |

Table 7: Performance comparison of the proposed and state-of-the-art methods on the IEEE PHM 2012 Prognostic challenge dataset (RMSE / Score)

| Dataset | Babu et al. 2016 | Hsu et al. 2018 | Wang et al. 2018 | Liu et al. 2019 | Proposed |
|---|---|---|---|---|---|
| 1_3 | 46/0.97 | 7579/100.65 | 7021/93.25 | 300/6.36 | 2/0.052 |
| 1_4 | 1690/372.54 | 5332/460.27 | 6556/471.65 | 2299/1059.8 | 1671/341.20 |
| 1_5 | 5159/341.55 | 27887/902.5 | 20635/667.9 | 8374/554.4 | 2360/156.25 |
| 1_6 | 935/223.98 | 6872/532.72 | 5329/413.11 | 1998/478.45 | 1227/293.86 |
| 1_7 | 26/6.58 | 1534/264.61 | 1185/204.47 | 0.742/3.41 | 0.213/0.027 |
| 2_3 | 1031/13.69 | 7000/92.96 | 933/12.39 | 33.21/0.441 | 0.963/0.013 |
| 2_4 | 365/26.29 | 5330/457.74 | 4496/323.49 | 1570/112.98 | 309/22.24 |
| 2_5 | 4226/144.86 | 20635/667.8 | 10200/330.1 | 6723/217.59 | 1099/35.59 |
| 2_6 | 123/9.59 | 5329/413.11 | 3478/269.68 | 1658/326.73 | 236/18.34 |
| 2_7 | 127/24.23 | 1185/204.47 | 803/138.01 | 327/216.12 | 74/21.93 |
| 3_3 | 1816/221.49 | 1865/227.14 | 1715/209.26 | 1927/235.01 | 631/77.03 |