# OpenReview forum: "Parallel Attention Network using Vector with High Correlation with Label for Remaining Useful Life Estimation"
_AAAI.org/2022/Workshop/ADAM — AAAI 2022 Workshop ADAM_

### Official Review · Reviewer_2z9F · 2021-11-29
**Spatiotemporal attention mechanism for RUL estimation**

**Rating:** 6
**Confidence:** 4

**Review:**

The paper proposes a deep learning architecture involving spatial (along the feature dimension) and temporal attention blocks to predict remaining useful life (RUL) for engineering systems. The paper validates the proposed framework with a well-known use case in the PHM community, C-MAPSS engine health monitoring data set. Empirical results show marginal performance improvement over state-of-the-art RUL estimation methods. The relevance of the paper in this workshop is a bit low. However, spatiotemporal attention based (interpretable) models are generally useful for engineering system modeling. While the paper reviews literature relevant to the general concept of attention mechanism, there have been many works recently in the area of spatiotemporal attention for multivariate time series analysis (e.g., Gangopadhyay, Tryambak, et al. "Spatiotemporal attention for multivariate time series prediction and interpretation." ICASSP 2021-2021 IEEE International Conference on Acoustics, Speech and Signal Processing (ICASSP). IEEE, 2021.) which are not discussed adequately. A brief discussion comparing and contrasting the proposed framework with other existing spatiotemporal attention architectures will be useful.

---

### Official Review · Reviewer_9hJ5 · 2021-12-01
**Good paper**

**Rating:** 6
**Confidence:** 3

**Review:**

The authors present an attention based architecture to predict Remaining Useful Life (RUL) for engineering systems. The proposed model consists of three paths -- a time-oriented attention network, a feature-oriented attention network, and a bidirectional long short-term memory (LSTM) network.

Pros:
1. The proposed model performs better than other approaches used for RUL prediction on the dataset analysed.
2. The architecture is well motivated for the problem. The authors also show ablation studies to motivate the choice of the various subcomponents.

Cons:
1. The approach is tested only on a single dataset (C-MAPPs turbofan engine RUL prediction). This, however,  seems to be the standard for other methods solving this task.
2. The approach is almost a straightforward application of existing attention networks. Therefore, there is limited novelty in the work.